# Balancing Freshness and Sustainability: Charting a Course for Meat Industry Innovation and Consumer Acceptance

**DOI:** 10.3390/foods13071092

**Published:** 2024-04-02

**Authors:** Emanuele Spada, Rachele De Cianni, Giuseppe Di Vita, Teresina Mancuso

**Affiliations:** 1Department of Agriculture (AGRARIA), University Mediterranea of Reggio Calabria, Feo di Vito, 89124 Reggio Calabria, Italy; emanuele.spada@unirc.it; 2Department of Agricultural, Forest and Food Science (DISAFA), University of Turin, Largo Braccini, 2, 10095 Grugliasco, Italy; rachele.decianni@unito.it (R.D.C.); teresina.mancuso@unito.it (T.M.); 3Department of Agriculture Food and Environment (Di3A), University of Catania, Via S. Sofia 100, 95123 Catania, Italy

**Keywords:** intelligent and active packaging, food industry, sustainable packaging, acceptance of technology, consumer preferences, consumer behavior

## Abstract

The agribusiness sector is constantly seeking solutions to enhance food security, sustainability, and resilience. Recent estimates indicate that one-third of the total food production remains unused due to waste or limited shelf life, resulting in negative environmental and ethical consequences. Consequently, exploring technological solutions to extend the shelf life of food products could be a crucial option to address this issue. However, the success of these technological solutions is closely linked to the perception of the end-consumers, particularly in the short term. Based on these considerations, this paper presents a systematic literature review of the main technological innovations in the fresh meat industry and of consumers’ perceptions of such innovations. Regarding innovative technologies, this review focused on active and smart packaging. Amidst various technological innovations, including the utilization of fundamental matrices and natural additives, a noticeable gap exists in consumer perception studies. This study represents the first comprehensive compilation of research on consumers’ perceptions and acceptance of innovations designed to extend the shelf life of fresh meat. Moreover, it sheds light on the existing barriers that hinder the complete embrace of these innovations.

## 1. Introduction

There is currently a growing demand for high-quality meat products, accompanied by increasing concerns about food waste and the environmental footprint of the agribusiness sector [1]. Therefore, preserving meat freshness is of the utmost importance. However, this issue is not isolated; the problem of food safety is also gaining prominence as microbiological hazards in food products continue to be a significant cause of foodborne illnesses [2]. In fact, according to the World Health Organization, approximately 420,000 deaths caused by unsafe food occur each year [3]. One of the primary reasons for food spoilage is the proliferation of microorganisms, which can originate from within the food or from external contaminants. These microorganisms produce undesirable byproducts that render the food unhealthy and unsuitable for human consumption [4].

Concerns about meat food safety stem from changes in animal production and distribution, increased international trade, and consumer preference for semi-processed products [5,6,7]. Often, improper management leads to food contamination of meat or meat products, which can occur during distribution, processing, catering, or retailing [8]. Such contamination can lead to the spread of foodborne outbreaks, with meat and meat products being a major source [9]. The consequences of even one case of a foodborne outbreak are significant and can have far-reaching impacts on businesses, as well as loss of life [10]. In Warmate’s study [11], foodborne illnesses are reported with data from the United States alone, showing that, from 2007 to 2012, 163 outbreaks were identified, with 4132 illnesses, 772 hospitalizations, and 19 deaths associated with meat and poultry.

It is also crucial to address the staggering level of food wastage. According to estimates from the United Nations Environment Program (UNEP) and the Food and Agriculture Organization (FAO) in 2022, approximately one-third of the total food production goes unused [12]. This wastage is attributed to factors such as shelf-life limitations, resulting in significant environmental and ethical repercussions.

To tackle these intertwined issues, the food industry has increasingly focused on developing innovative methodologies to monitor and maintain food quality and freshness, as highlighted in recent research [13]. Examples include the use of smart labels or packaging with the release of additives that allow the shelf life of products to be increased [14]. Moreover, the meat sector has emerged as a primary area of innovation, considering the anticipated surge in meat consumption. Recent studies have highlighted how meat availability and consumption are expected to increase by 5.9 and 14 percent, respectively, by 2030 [15]. This notable increase can be attributed to rising household incomes, especially in developing Asian countries, and to the growth of the world’s population. Considering these growth trends and the aforementioned risks, an overall enhancement of meat quality, particularly in terms of extending its shelf life, is necessary.

Along with these considerations, it is increasingly important to prioritize the use of environmentally sustainable solutions, aligning with government policies in Europe and many other countries around the world. To date, traditional packaging, derived from petroleum, has been complemented by biobased packaging derived from renewable sources such as biomass, microorganisms, and biotechnological sources [16], with the latter source experiencing growing usage. In Europe, this approach is further encouraged through pro-sustainability policies like the Green Deal, which aims to achieve climate neutrality by 2050 through resource-efficient strategies. This includes transitioning to a clean, circular economy to restore biodiversity and reduce pollution. These goals align with the potential to reduce greenhouse gases by 40 percent from the levels of 1990, which could be facilitated by reducing the use of petrochemicals, particularly in packaging.

Within this context, this study examines various potential differentiation strategies that industry players could implement on a large scale to address these challenges, while also taking into account consumer preferences and acceptance. This paper presents a pioneering systematic literature review that uniquely combines both technological and economic aspects. It offers a comprehensive exploration of how the meat industry can navigate the delicate balance between ensuring freshness and promoting sustainability, all while considering consumer acceptance. Through a multidimensional approach, this review sheds light on the intricate interplay between technological advancements and economic viability within the meat industry landscape.

To this end, this systematic literature review highlights the advancements made in packaging technologies within the meat industry, such as active packaging and intelligent labels. These technologies not only improve meat preservation but also align with sustainability goals. Until recently, the terms “active” and “intelligent packaging” were often used interchangeably [17]. However, according to Htnu et al. [18], there is now a clear distinction between these terms, as follows: active packaging refers to technologies that enhance the safety and preservation of products, while smart packaging refers to technologies that transparently communicate useful product information in real time to various actors in the supply chain. Additionally, innovative processing techniques, including advanced meat treatments with natural antimicrobial agents, are discussed herein.

The structure of this study is as follows: after a general description of the article selection method, the theoretical background and research hypotheses are presented. Subsequently, the results on technological innovations related to extending the shelf life of products and smart labels, as well as studies on consumer perceptions of such innovations, are outlined.

## 2. Conceptual Background and Research Hypothesis

Firms in a highly competitive business landscape seek to gain a competitive edge over their competitors through technological product and process differentiation [19]. Technological product differentiation involves creating unique and innovative products that stand out on the market, as they provide distinct features, functionalities, and/or benefits to customers [20]. This differentiation can be achieved through advancements in product design, materials, and/or technological components. Similarly, technological process differentiation involves implementing efficient and advanced manufacturing or operational processes that enhance productivity, quality, cost-effectiveness, and/or health-related attributes [21,22,23]. By adopting cutting-edge technologies, firms can streamline their operations, improve efficiency, and deliver superior products or services to customers [24,25].

Indeed, the validation of technological innovations is closely linked to consumers’ perceptions. Only after investigating their propensity to buy and willingness to pay for these innovative solutions can such products be launched on an industrial scale. By combining more sustainable approaches with consumer acceptance, the industry can effectively increase the shelf life of fresh meat, while minimizing waste and reducing the environmental impact associated with meat production and distribution.

Consumers today demand meat products that are safe, healthy, high quality, and convenient to purchase and use [26], and food industries strive to meet these demands. However, as mentioned earlier, this condition leads to significant food waste. According to some authors, extending the shelf life of meat products, even by just a couple of days, can substantially reduce food waste in households, resulting in benefits for both spending and the environment [27].

Most review studies conducted to date have primarily focused on technological innovations aimed at extending the shelf life of various food products in general. Examples include the works of Fadiji et al. [28], Echegaray et al. [29], and Khodaei et al. [30]. Some other review studies have specifically addressed innovations in fresh meat and meat products, such as those by Gil and Rudy [31] and Zhang et al. [32].

To the best knowledge of the authors, this is the first study in which a systematic literature review has been conducted, combining innovations with consumers’ acceptance. This research investigates the significance of technological innovations as a crucial strategy for firms aiming to effectively differentiate their products, along with discussing consumers’ perceptions associated with this strategy. Based on these premises, this study underscores the importance of consumers’ acceptance in the implementation of these innovative solutions.

**H1:** 
*What are consumers’ perceptions of innovative packaging for fresh meat?*


## 3. Materials and Methods

This research aims to investigate innovative solutions adopted within the meat industry to extend the shelf life of fresh meat, while also integrating a sustainable approach and ensuring consumers’ acceptance. To achieve this goal, the Preferred Reporting Items for Systematic Reviews and Meta-Analyses (PRISMA) methodology, as outlined in the papers by De Cianni et al. [33] and Stillitano et al. [34], was followed. Accordingly, a bibliographic search was conducted in scientific article databases such as Scopus and Web of Science (WOS).The combination of the main keywords, using AND/OR Boolean operators, were as follows: (“food” OR “agr*” AND “meat” AND “packaging” AND “consum*” OR “innovati*” OR “consum* behav*” OR “consum* preference*” OR “consum* attitude” OR “consum* concern*” OR “consume* intention*”), as reported in Table 1.

The databases were consulted in July 2023, and the searches conducted in the Scopus and Web of Science (WOS) databases yielded 1356 and 598 articles, respectively, totaling 1954 papers (Figure 1). Duplicate papers were excluded and then subjected to a screening process. Initially, the “Refine Results” tool of the databases was utilized to exclude reviews, book chapters, and editorial materials, as well as studies published before the year 2004, while only papers written in English were included. Consequently, only indexed application references were taken into account.

A second screening was conducted based on the content of the abstracts, during which discussion articles, off-topic studies, and those not centered on innovations in fresh meat, or consumers, were excluded. Through this process, 164 articles were assessed for eligibility via thorough examination of the full texts, with studies not directly pertinent to the research topic being discarded. Consequently, the total number of articles was reduced to a final selection of 116 representative papers, which were included in the qualitative synthesis. Each of these articles was read in full and analyzed individually for the purposes of this study. Additionally, papers that were not obtained directly from the initial search with the specified keywords but were referenced in the analyzed papers were also included in this comprehensive review.

## 4. Results and Discussion

### 4.1. Descriptive Statistics

Out of the 116 papers, 96 were identified regarding technological innovations in fresh meat (the other 20 papers refer to consumer studies), with some papers discussing more than one type of meat (Figure 2), The types of meat investigated included chicken (n = 33), beef (n = 28), pork (n = 19), as well as other types such as turkey, buffalo, bison (n = 10), and meat products (n = 10). Chicken meat was the most commonly studied, likely due to its status as the most produced meat globally [35] and its association with foodborne illnesses resulting from contamination [36].

As depicted in Figure 3, the temporal distribution of articles investigating industrial innovations shows that the majority of them were published in 2022 (n = 18), followed by 2021 (n = 12), and 2018 (n = 10). There has been a notable increase in publications since 2019, indicating a growing interest in the topic. Out of a total of 20 articles focusing on consumer perceptions, sixteen examined the perceptions of innovations related to shelf-life extension, while four focused on intelligent packaging. Regarding the papers on extending shelf life, six investigated beef, generic meat, and meat products, two focused on lamb, two on chicken, one on pork, and one on generic food. These studies were conducted in various countries, including China, Poland, Spain, France, Germany, the United Kingdom, Italy, Australia, the United States, Canada, Iceland, and Brazil. In almost all of these countries, in particular in China, Brazil, Spain, Mexico, India, Canada, and the USA, the amount of meat produced show a tremendous increase [34]. Moreover, in many of these countries, per capita consumption is also often double the world average [37].

### 4.2. Technological Innovations in Meat Industry Packaging

In this section of the study, the technological innovations in meat industry packaging (96 findings) were categorized into the following groups:(a)*Active packaging*, characterized by the composition of the primary biodegradable matrix and the incorporation of active compounds.(b)*Product-centered innovations*, involving the direct application of active substances or external treatments to the product.(c)*Intelligent packaging*.

The innovations associated with active packaging primarily focused on the materials used for packaging. Studies examined various materials, including cellulose [15,38,39,40,41], chitosan [42,43,44], starch [45], natural gums [46], whey protein [47,48], alginates [49], glycerol [50], and polylactic acid [51,52].

These packages incorporate additives with various functions, including antimicrobial properties, which can prolong the shelf life of a product by inhibiting microbial growth through the release and diffusion of these compounds from the packaging to the food surface [53,54]. Additionally, they may serve antioxidant and antibacterial functions. The exploration of natural substances for inclusion in meat packaging has led to the identification of several compounds, driving innovation in the meat industry. Natural additives added to fresh meat packages include quercetin [55], garlic extracts [5], curcumin [45], lysozyme [56], lemongrass essential oil [44], ozonated olive oil [45], feijoa [56], and persicara minor [50].

Their antimicrobial action is highlighted by several studies as in that Dirpan et al. [15] in wich demonstrated the antimicrobial effects of adding 10–15% garlic extract to beef packaging, while Tsironi et al. [47] explored the use of ginger and rosemary essential oils on lamb. Antibacterial action was observed by Ali et al. [42] through the use of pectin and chitosan-based nanoactive films against multidrug-resistant meat pathogens such as Salmonella enterica, Escherichia coli, and Listeria monocytogenes. Contini et al. [43] reported excellent results against *Escherichia coli* and *Staphylococcus aureus* using chitosan films activated with lemongrass essential oil and Nazari et al. [57] incorporated essential oils of Ziziphora clinopodioides into the film. Sganzerla et al. [58] demonstrated efficacy against *Escherichia coli*, *Salmonella*, and *Shigella* using starch, citric acid, and functionalized pectin with feijoa, while Hu X et al. [56] utilized an edible cellulose film activated with lysozymes. Tan et al. [59] showed that adding Chrysanthemum morifolium to a 1.5 percent (*w*/*v*) chitosan film increased the shelf life of raw meat products by significantly reducing Staphylococcus aureus activity. Additionally, antioxidant activity was observed in studies by Guo et al. [49], who incorporated carboxylated cellulose nanocrystals into films, and Ahmad et al. [60], who treated meat with kinnow and apple peel powder.

When it comes to product applications, innovations in the industry can be categorized into direct applications and coatings with edible substances.

Direct applications involve incorporating natural antioxidants into meat products during their preparation. This can be achieved by spraying an antimicrobial solution onto the surface or soaking the meat in such a solution [61]. Various types of antimicrobial additives used for this purpose include apple and citrus peel [60], rosemary and oregano essential oils [62], lemon verbena and clove essential oils [61], hyssop and coriander essential oils [63], and lactic acid [64].

Edible coatings, composed of polysaccharides, proteins, and lipids, also play a role in extending the shelf life of food products by creating a barrier against gases, solutes, and vapors [65]. Moreover, antimicrobial compounds can be incorporated into these coatings to maintain high preservative concentrations on food surfaces, similar to those observed in packaging [66]. Examples of edible coatings aimed at extending shelf life include those made with chitosan and gallic acid [67], chitosan and sodium alginate [68], soy-based protein isolates [69,70], gelatin [71,72], and apple fruit extracts [73].

Regarding the use of radiant energy for food sterilization and preservation, research has shown excellent inactivation results of microorganisms even at low energy levels [74,75]. In addition, the use of radiant energy, within some limitations, does not alter the main sensory characteristics of meat, such as color, flavor, and texture [76].

Other emerging technological treatments used to extend the shelf life of meat include low-energy electron beam [77], atmospheric cold plasma [78,79], moderate electric field [80], gamma radiation [81], and ohmic heating [82]. The innovative “intelligent packaging” technology serves the primary function of providing indications on the preservation status of fresh meat, thereby enhancing food safety for consumers [83]. This information is visually communicated to consumers through changes in the color of a plastic film or specific labels placed inside a package [84]. The color change and related information are associated with microbial growth, chemical reactions, time–temperature variations, freshness, and packaging damage [85]. Such packaging can be produced from natural matrices to which equally natural additives are incorporated, including anthocyanins from potatoes [86] and curcumin [87].

### 4.3. Consumers’ Acceptance

Although there is extensive research on technological innovations, as highlighted above, there is a lack of studies specifically focused on consumers’ perceptions. This gap is particularly noticeable in emerging countries, as noted in the study by Barone et al. [88] and supported by Wang et al. [71].

After providing a general overview of the methodological approaches used in the 20 consumer-related papers, the results have been categorized into two sections, as follows: acceptance of innovations related to extending shelf life and acceptance of innovations related to smart labels.

The reviewed studies covered a wide range of research topics and employed various methodological approaches. Table 2 illustrates the methodological approaches adopted in the research papers. An acceptance analysis was conducted in six papers, choice experiments including willingness-to-pay assessments were conducted in seven papers, qualitative assessments were conducted in five papers, and a single paper each investigated willingness to buy and performed a correspondence analysis.

Upon analyzing the literature, interesting results emerge from a comprehensive exploration of the methodologies adopted in consumer perception studies, encompassing both quantitative and qualitative approaches. This analysis sheds light on the limitations or biases that could impact the interpretation of results or the validity of any approaches.

Regarding choice experiment-based methodologies, Ardeshiri et al. [97] report that the limited graphical space for the attributes analyzed in the choice experiment task may represent a restriction, as individuals may not evaluate all information when making their decision. Similarly, limitations are apparent in the study by Wang et al. [71], where the random draw method used to generate different sets of choices may be optimal according to various efficiency criteria, but may not offer the highest efficiency compared to the D-optimal or fractional factorial design. Other limitations were identified in the study by Erdem [98], who conducted a discrete choice experiment and analyzed stated preference choice using a multinomial logit model. This study revealed that a limitation of the analyzed model is the assumption of homogeneous preferences for all respondents.

Conversely, Chen et al. [96] use a non-hypothetical choice experiment to isolate the impact of individual product characteristics on price and consumer choice decisions. Such choice experiments are preferable to hypothetical experiments as they mitigate the potential hypothetical bias in participants’ choice decisions.

In order to report results in a manner as objective as possible, Wang et al. [89] eliminated questionnaires completed in less than one-third of the average time and those containing seven or more identical answers. In fact, improbable response patterns and sub-standard response times are signs of low-quality responses.

As for studies investigating consensus based on qualitative analysis, no particular limitations can be observed. From an operational point of view, many works conduct the investigation of consumer perceptions in several stages. In the study by Pennanen et al. [103], the operational phase is divided into three parts. The first part examines respondents’ knowledge about the meat product and the type of packaging, the second part provides participants with information about the time–temperature indicator (TTI) concept, and a third part in which the two commercially available TTI applications (TTI1 and TTI2) are presented and discussed.

Barone et al. [88] also divide the study into three phases. In the first phase, they analyze consumers’ general perceptions of freshness and expiry dates, as well as the purchasing, usage, storage, and disposal habits of red meat. The second phase analyzes consumers’ perceptions of smart labels and their effect on food handling practices. Finally, the third phase explores consumer perceptions of smart labels and how these could be integrated and influence consumer practices in relation to the use, purchase, storage, and disposal of red meat.

Htun et al. [18] also divided the study into three phases. In the first phase, the participants were informed about the project, together with other general information. In the second phase, the needs and desires of the participants were explored, based on the features and benefits of the information transmitted by the smart tags, and, finally, in the third phase, the acceptability of different smart tag solutions was investigated.

Table 3 illustrates the various data collection methods employed in the research studies. Among these methods, the questionnaire, often administered online, was the most commonly utilized, followed by focus groups and sensory evaluations. Interviews, open-ended surveys, and observations were less frequently employed.

#### 4.3.1. Innovations Related to Increasing Shelf Life

As highlighted by Wang [89], consumers’ acceptance of new technologies is not always certain and varies depending on numerous factors. The analysis of the perception of specific innovative technologies revealed that most studies focused on relatively simple and basic technologies. This trend may be attributed to the significant divergence in perceptions among countries, where even seemingly simple innovations are perceived as groundbreaking in some cases. One of the most commonly identified obstacles to consumer acceptance in these studies is the lack of consumer knowledge about these technologies. This lack of awareness often prevents consumers from forming objective judgments. Indeed, the articles underscore that these issues are still not widely understood among consumers but, when recognized, they tend to influence consumer choices.

Table 4 provides an overview of the characteristics of the articles included in this review that investigated consumers’ perceptions of technologies designed to extend the shelf life of fresh meat.

The study conducted by Stoma et al. [104] on a sample of Polish consumers revealed that both men and women paid close attention to visually appealing and environmentally friendly packaging, with recyclability and readability being key factors. Interestingly, there were only minor differences observed between male and female groups. However, the study also found that consumers, regardless of gender, showed limited interest in innovative solutions such as active and intelligent packaging, indicating a lack of awareness regarding the benefits and functionalities of such packaging.

In another study focusing on Chinese consumers, willingness-to-pay (WTP) was examined by analyzing attributes related to the storage and packaging of pork [71]. While this innovation may seem trivial, it is important to note that, in some regions of China, purchasing pork from markets lacking refrigeration facilities is still common practice. The study investigated packaging technologies such as vacuum pre-cut pork and plastic packaging as alternatives to bulk meat. Results indicated a strong preference for refrigerated and packaged meat among consumers. However, approximately half of the buyers continued to purchase unrefrigerated pork from wet markets. Concerns about the impact on meat quality and safety posed a barrier to purchasing chilled or frozen meat. Nevertheless, providing more detailed information about refrigeration led to a significant increase in consumers’ willingness to pay, indicating a shift in preferences and behaviors.

In another study involving American consumers and focusing on beef packaging, Ardeshiri et al. [97] found a preference for vacuum packaging over packaged trays.

Van Wezemael et al. [94] conducted a study with 2520 European consumers across France, Germany, Poland, Spain, and the United Kingdom, analyzing perceptions of various packaging innovations for beef. Vacuum packaging was the most widely accepted, with 73.0 percent approval, followed by modified atmosphere packaging at 54.7 percent. However, packaging technologies involving additives received less favorable responses, particularly those releasing food additives from the package, with over 40% of respondents finding this unacceptable. The study identified four distinct consumer profiles, as follows: enthusiastic, cautious, negative, and conservative.

Polkinghorne et al. [93] analyzed Australian consumers’ perceptions of three types of beef packaging, oxygen-permeable film packaging (OWP), vacuum packaging (VSP), and modified atmosphere packaging (MAP: 80% O_2_ and 20% CO_2_). While all three technologies aimed to increase shelf life, MAP technology showed reduced overall acceptability compared to OWP and VSP, with no substantial difference between the latter two.

Grebitus et al. [27] used choice experiments to assess American and German consumers’ preferences for modified atmosphere packaging (MAP) and carbon monoxide modified atmosphere packaging (CO-MAP) of meat. American consumers were initially unwilling to pay for shelf-life extensions without knowledge of the packaging technologies, but after being informed about MAP technology, they were willing to pay more for extended shelf life. German consumers did not show a significant positive response to longer shelf life, but some preferred products with longer shelf-life attributes. However, the introduction of CO-MAP had a negative effect on both American and German consumers’ preferences. Similarly, Guzek et al. [90] conducted a study among Polish consumers to analyze perceptions and acceptance of innovative meat product packaging, particularly when coupled with an overall improvement in quality.

Sodano et al. [100] investigated the perception of 300 Italian consumers regarding food nanotechnology, including antimicrobial packaging for meat. The study revealed a low level of consumer confidence in these innovations, with a perception of higher risks compared to expected benefits, coupled with a lack of knowledge about the technologies and a degree of food technophobia.

Erdem [98] analyzed the acceptability of nanotechnology in packaging, focusing on chicken meat, among UK consumers. The study found that consumers primarily focused on food poisoning risks. Although more than half of the consumers (51%) viewed the use of nanosensors in chicken packaging as a good idea, there were varying levels of concern among respondents. However, the majority of consumers were willing to choose chicken with innovative technology over the status quo alternative. Despite concerns about food risks, consumers did not exhibit strong preferences between nanosensors and conventional methods, but they were willing to pay more for chicken with nanosensors. Additionally, consumers who purchased chicken with nanosensors were more concerned about animal welfare compared to consumers of conventional chicken. Overall, consumers preferred chicken with lower food poisoning risks, better animal welfare, and reasonable costs, regardless of the presence of nanosensors.

Extending the discussion to various innovative technologies for fresh meat packaging, Wang K. [89] investigated Chinese consumers’ perceptions of meat products using the Likert acceptance scale. The perception assessment occurred in three stages, as follows: first, based on their prior knowledge; second, after providing information related to L. monocytogenes; and, finally, after providing information related to the specific technology in question. Consistently, consumers’ perception improved after receiving information on all the analyzed technologies. In the initial condition without additional information, the most accepted technology was traditional thermal pasteurization (TP), while the least accepted were bacteriophages (BPs) and rinsing meat carcasses with antimicrobial solutions (RMCA). Interestingly, antimicrobial packaging (AP), despite the term “antimicrobial,” was still more readily accepted than RMCA. Although both AP and RMCA involve antimicrobial compounds, their acceptance differed, with 94.2% accepting AP. However, the acceptance of TP and high-pressure processing (HPP) technologies dramatically decreased in the second evaluation, with increased acceptance of BP. This shift could be attributed to heightened awareness of the risk posed by L. monocytogenes, prompting respondents to reconsider the suitability of traditional technologies for its control. In the third assessment, the two most acceptable technologies remained TP and HPP, while RMCA remained the least acceptable.

Shifting the focus from packaging to other types of innovations, Johnson et al. [91] studied the acceptability of ready-to-eat meat products for American chicken sausages and cubed meat irradiated with an electron beam at 1, 2, and 3 kGy. The study demonstrated that the irradiation technology used had no negative effects on consumers’ acceptance or on other sensory characteristics. In terms of shelf life, both frankfurters and chicken meat were accepted for longer durations, with samples lasting 32 and 18 days, respectively, compared to non-irradiated samples.

In another study, Horrillo et al. [102] investigated Spanish consumers’ perceptions of adding natural ingredients, such as cherries and pecans, to lamb burgers to enhance their nutritional properties and preservation. Despite lamb being consumed less frequently than other meats, participants considered meat obtained from pasture-raised animals as indicative of healthier options. Overall, consumers accepted the proposed innovation, irrespective of their liking for lamb, as the added compounds masked the lamb flavor.

Mauricio et al. [92] focused on lamb as well, examining Brazilian consumers’ perception of chitosan-coated meat. The results revealed that samples labeled with ‘edible chitosan coating’ without further explanation had a positive effect. However, when the label mentioned ‘technology to preserve meat properties’, a negative utility value was observed. This discrepancy might be attributed to the association of chitosan with non-natural additives, like synthetic preservatives, which have recently been linked to health and safety concerns, such as carcinogenic and toxicological effects. This study emphasized the importance of providing consumers with comprehensive product information to prevent associations with potentially harmful additives.

#### 4.3.2. Innovations-Related Intelligent Labels

This sub-section presents the selected articles related to the study of consumers’ perception concerning innovations in intelligent packaging for fresh meat (Table 5).

Htun et al. [18] conducted an analysis on the opinions of focus groups in the UK, Finland, Spain, Poland, and Iceland to formulate hypotheses regarding future packaging. The results of the study showed that food safety was a significant concern that influenced consumers’ choices. In terms of innovation, freshness indicators were favorably perceived by the respondents as they enhance the informational value of packaging. However, it is crucial that these indicators should be associated with the consumer’s “common sense”. Notably, when making choices, the freshness indicators were consistently associated with other features, such as QR codes. On the other hand, when assessing the value added by innovations, freshness information was linked more closely to the entire food supply chain, product quality, and health-related information. Despite the concern for the environment, recycling was ranked as one of the last issues in order of importance. However, in terms of willingness-to-pay, consumers were only willing to pay a premium price for food products if the higher prices could be justified by the increased costs for producers to make their operations more environmentally friendly and sustainable.

Food safety has also been found to influence consumers’ perceptions of innovation. In the work of Nocella et al. [99], the WTP for innovation related to smart biosensors was analyzed considering the perceived risk related to food safety. The proposed food risk was that caused by Escherichia coli in meat products. This is an interesting result because it indicates that the information provided during food safety alerts mainly triggers an emotional reaction (fear) and a cognitive elements of threat assessment. The protection motivation theory (PMT) was used in the study to explain WTP and consumers’ acceptance regarding the use of biosensors for meat products observed in different risk communication scenarios. PMT elements seemed to influence consumers’ WTP for smart biosensors in opposite ways in the two analyzed scenarios. In a risk-free situation, consumers’ WTP was influenced by a coping assessment but not by fear, while the opposite result was observed in the risk scenario. The study proposed two groups, “In the absence of risk” (NRI) and “In the presence of risk” (RI). The results regarding the willingness to purchase meat with biosensors showed a high propensity to purchase in both groups (NRI = 83.3%, RI = 84.4%). These results indicate that consumers’ adoption of protective behaviors was relatively in line with the acceptance of innovation. However, there was a higher average WTP (£0.91, s = £0.72) in the RI group than in the NRI group (£0.82, s = £0.68). This result suggests that although innovation had a positive effect on consumers, information provision and increased risk awareness increased the WTP. This was a consequence of the consumers’ perceived effectiveness of biosensors in reducing the risk of food poisoning. The fear component had no impact on WTP when risk information was not provided. However, price mediates protective behavior and it was observed that the estimated surcharges for biosensors showed negative peaks for the NRI group, while the surcharges for meat packed with biosensors showed a clear positive peak for the RI group.

There is a growing trend in the realm of fresh meat packaging involving the use of smart labels. These innovative labels offer consumers valuable information, and understanding consumers’ perceptions of these labels is crucial for both producers and consumers alike. In this direction, Barone et al. [88] analyzed the perceptions of British consumers regarding smart labels applied to fresh meat packages. The results of the study showed a high interest of the participants in trying this type of innovation, both in the store and at home. The smart label was perceived as a reliable guarantee of freshness beyond expiration dates and personal perceivability. In addition, it was considered that this innovation would lead to a reduction in food waste, as it could help consumers consume all the food before disposing of it, as well as eliminating the inconvenience and fear of contamination associated with damaged food.

Furthermore, smart labels were linked to a higher propensity for packaging recycling, thereby being aligned with more sustainable practices. However, concerns emerged in all the groups related to the lack of accurate knowledge of the inner workings of the smart label, increased price, and the environmental impact of an additional label on the package [88].

It was also found, in the context of meat labeling, that consumers did not immediately rely on the label but referred to the sensory characteristics of the product, especially for products that had passed their expiry date. Pennanen [103] analyzed consumers’ perceptions of a time–temperature indicator (TTI) that provided information on temperature fluctuations in food products, including meat. The study was conducted through a quantitative survey on 16 focus groups in Finland, Greece, France, and Germany. It was found that the adoption of TTI-related innovations on the consumer market was still low, despite the many benefits available for all stakeholders. The results of the study showed that, in Greece, France, and Germany, the TTI technology was seen as synonymous with safety and security, as a result of the increased belief in cold chain compliance, improved product handling and storage, and increased transparency. In Greece and France, TTI was perceived to instill consumer confidence, as a result of improved food handling and storage, as well as cold chain management before purchasing a product. In Greece, Germany, and Finland, consumers attributed TTIs to supporting the purchasing of such products by serving as additional selection criteria. In addition, especially in Germany, it was found that innovative technology may be especially supportive to older people who have difficulty reading labels. A comparison between countries showed that Finnish consumers considered the benefits of TTIs to be less important than in other countries, while the overall evaluation of TTIs was significantly higher in Greece. However, technology-related concerns also emerged, such as the one related to the potential price increase highlighted by German, Greek, and Finnish consumers. In fact, especially in hot countries, for example in parts of France, consumers tended to attach more importance to the sensory perception of food than consumers in other studied countries. Different climatic conditions in different countries affected the acceptance of TTI. In the south of France and in Greece, due to the high summer temperatures, which lead to a greater deterioration of products due to incorrect procedures in the cold chain, the acceptance of TTI innovations was higher. Therefore, consumers’ perception of this type of innovation was influenced, in particular, by external conditions such as the country in which one lives.

#### 4.3.3. Drivers Related to Consumers’ Acceptance

Several recurring factors have been identified as key drivers in the various studies that examined consumers’ acceptance of innovative and sustainable solutions designed to extend the shelf life of fresh meat. These factors included gender, household composition, income, age, education level, place of residence, and product price.

Regarding the perception of smart labels, no major gender differences were found in the study by Barone et al. [88]. However, in Pennanen’s study, conducted in France and Germany [103], it emerged that female participants associated more benefits with time–temperature indicators (TTIs) than men. This difference was not found in Greece and Finland.

Another important socio-economic aspect, highlighted in two different studies conducted in Greece and Finland, was the composition of the household. In Greece, respondents belonging to households with children perceived the TTI as more beneficial than those belonging to single households, probably due to food security reasons. In Finland, respondents from households with other adults differed in perceiving the TTI as more advantageous, showing a higher preference among consumers with a lower average age.

In terms of income, respondents with higher earnings were found to be more willing to accept innovations [89,92,96]. According to these authors, the most influential determinants of acceptance for this category of consumers were “production technology”, “expiry date”, and “cost” [90]. Conversely, in a study conducted by Wang H. et al. [71], it emerged that income did not influence the consumers’ behavior concerning this type of choice.

With regard to age, in both Wang K.’s [89] and Stoma’s [104] studies, it emerged that the consumer groups that accepted innovations most willingly were those included in the following categories: 25–34 years old and 25–45 years old. However, in Stoma’s study [104], it emerged that awareness of this topic was quite low for the under 25 and over 60 age groups. This was also confirmed in a study by Wang H. [71], where older consumers showed reticence toward innovations.

It was observed in previous studies that high levels of education also positively influenced the acceptance of innovative and sustainable solutions to extend the shelf life of fresh meat through several mechanisms [71,89,94,96].

Firstly, individuals with higher education levels often have greater exposure to information and awareness about food safety, preservation techniques, and environmental sustainability. This knowledge allows them to better understand the benefits of sustainable meat preservation methods, such as reduced food waste and the potential to lower environmental impacts [105].

Secondly, educated consumers tend to be more open to adopting new technologies and innovations. They are accustomed to critical thinking and assessing the scientific evidence behind novel approaches [106]. When presented with evidence of the effectiveness and safety of innovative meat preservation methods, highly educated individuals are more likely to embrace these solutions [107].

Furthermore, education fosters a greater appreciation for the long-term consequences of consumption choices. Consumers with higher education levels may be more forward-thinking and willing to invest in sustainable practices that offer benefits not only for themselves but also for future generations [105].

In summary, high levels of education can enhance consumers’ understanding of, receptivity to, and appreciation of innovative and sustainable solutions aimed at prolonging the shelf life of fresh meat. This educational influence contributes to the positive acceptance of such advancements in the food industry.

In Guzek’s study [90], influential determinants for this category of consumers included “Production technology” (process of production, certified production, and technology), “Manufacturer” (trademark, company producer), and “Components” and “Nutritional value” (minerals, vitamins, and food additives). Place of residence also appeared to influence consumers’ acceptance of innovation in meat products. In Stoma’s study [104], consumers residing in rural areas were more interested in the external appearance of packaging than in innovative solutions. Guzek’s study [90] found that consumers residing in small urban areas with a population of less than 100,000 paid more attention to production technology, cost, and manufacturer. In contrast, for consumers living in cities with more than 100,000 inhabitants, the most important attributes were the expiry date, manufacturer, and cost.

Finally, among the other variables that influence consumer acceptance, price plays a pivotal role. Chen [96] found that a lower price favored the purchase of meat subjected to innovative technologies, such as vacuum packing. Stoma et al. and Pennanen et al. argued that the higher price associated with innovations was a concern for the interviewed consumers [103,104].

## 5. Conclusions

The current study underscores the existence of numerous innovative technologies aimed at extending the shelf life of fresh meat. Moreover, there is a plethora of natural and renewable substances available for use in packaging or as bioactive compounds, aligning well with the objectives outlined in European Green Deal policies.

Additionally, this research emphasizes the necessity for continual investments in research and development, ongoing monitoring of technological advancements, and effective intellectual property protection to maintain and enhance the competitive edge derived from technological differentiation. Furthermore, the findings underscore the importance of technological product and process differentiation as crucial strategies for firms to prosper in a dynamic business landscape.

Regarding consumers’ acceptance, the results reveal disparities in the intensity of study across different technologies, particularly the newer ones. It can be inferred that while numerous innovations are reported in the literature, only a fraction have been thoroughly examined in terms of consumers’ perceptions. However, varying levels of innovation exist across different regions of the world. For instance, while purchasing meat in wet markets remains common in some parts of China, other regions exhibit a more proactive approach towards investigating innovations.

Addressing the initial question posed at the outset of this review—"What is the consumers’ perception of innovative packaging for fresh meat?”—the study suggests that no single answer suffices. Among the conducted studies, it becomes evident that not all innovations garner equal acceptance, with a notable preference for more “familiar” technologies observed among consumers.

Overall, consumers have shown reluctance to pay for shelf-life extensions when they lack information about the underlying packaging technologies. Furthermore, the acceptance of innovative technologies, such as antimicrobial packaging, is closely tied to consumers’ understanding of the technology and the associated risks of food contamination. Another significant barrier to acceptance is the perception of innovations solely as the addition of food preservatives, which is generally viewed negatively.

It can be inferred that, in many cases, consumers did not perceive innovations positively. In fact, in some instances, the provision of information about the technology or food risks even led to a loss of consumer confidence. For instance, regarding smart labels, when consumers associated them with perceived risks, there was a higher willingness to pay (WTP) for these innovations, as evidenced in the study by Lai et al. [108]. Smart labels were perceived as providing objective support, thereby reducing subjectivity regarding perceived risks based on scientific evidence.

Thus, it is evident that detailed, product-specific studies are essential to assist entrepreneurs in making informed industrial decisions and addressing consumer concerns effectively.

Overall, consumers have generally been reluctant to pay for shelf-life extensions when they lack information about the underlying packaging technologies. Additionally, the acceptance of innovative technologies, such as antimicrobial packaging, is closely tied to consumers’ understanding of the technology itself and the risks associated with food contamination.

Furthermore, the tendency to equate innovations with the addition of food preservatives represents a significant barrier that must be overcome [109], as it contributes to a purely negative perception of these advancements.

It can be inferred that, in many cases, consumers did not perceive innovations positively. However, in rare instances, information about the technology or food risks led to a loss of consumer confidence. Notably, in the case of smart labels, when associated with consumers’ perception of risk, a higher willingness to pay (WTP) for these innovations was observed, as confirmed in the study by Lai et al. [108]. Smart labels were perceived as providing objective support, thereby reducing the subjectivity of perceived risks based on scientific evidence. Thus, it is evident that detailed, product-specific studies are essential to assist entrepreneurs in making informed industrial choices and addressing consumer concerns effectively.

### 5.1. Implications

These findings carry significant implications for the food industry, guiding investment decisions towards innovations that are more readily accepted by consumers, while ensuring the economic sustainability of production processes. It is essential to recognize that price remains a critical factor influencing consumer choices [27]. However, introducing new technologies may pose challenges for existing food business models, adding technical complexity and requiring substantial investments [18].

Moreover, the development and implementation of innovations in the food industry are closely intertwined with government regulations, particularly concerning food safety. Therefore, ongoing collaboration between industry stakeholders and governmental bodies is imperative [110]. Government policies play a crucial role in shaping consumer behavior by providing clear public health messages and information. Such messages have the power to promote healthier products and influence consumer choices positively [111]. Research indicates that public health messages increase awareness and knowledge, leading to changes in behavior [112]. Many of the participants in the reported surveys showed a lack of knowledge and/or interest in information about traceability and the benefits of healthy eating, which could be a significant barrier to the development of the innovative solutions covered in this study. This barrier could be overcome through consumer education through the joint action of retailers and policy makers [29]. From an environmental sustainability perspective, active packaging is found to be more sustainable than conventional packaging, with lower rates of harm with respect to human health, ecosystems, and resources [113]. This condition is fully in line with European and global policies that are increasingly focused on the environmental and social sustainability of processes and products. Additionally, consumers are becoming more attentive to reducing environmental impact and mitigating the effects of climate change [114,115,116]. Therefore, a thoughtful implementation of these concepts could further motivate consumers to purchase products featuring innovations such as active packaging or intelligent labels.

### 5.2. Limitations of Study and Further Research

This study, while offering valuable insights, is subject to several limitations. Firstly, despite our best efforts to ensure the thoroughness of our review, it is possible that some aspects of the topic were not covered comprehensively, due to the limited availability of both quantitative and qualitative studies examining consumers’ choices related to meat consumption. Additionally, the vast array of innovative technologies in the meat industry presents a challenge in fully understanding consumers’ reactions. Finally, the sheer volume of available technologies can make it challenging to fully assess their impact.

Moving forward, it is clear that an integrated approach involving all stakeholders in the supply chain is necessary to develop effective solutions for improving food safety and reducing food waste. Researchers should continue to investigate innovations and consumers’ perceptions to guide food industries. However, these efforts must be supported by specific government initiatives aimed at safeguarding public health and promoting sustainable development policies, free from economic interests. Such initiatives are crucial for ensuring the health and well-being of communities and the environment in the long term.

## Figures and Tables

**Figure 1 foods-13-01092-f001:**
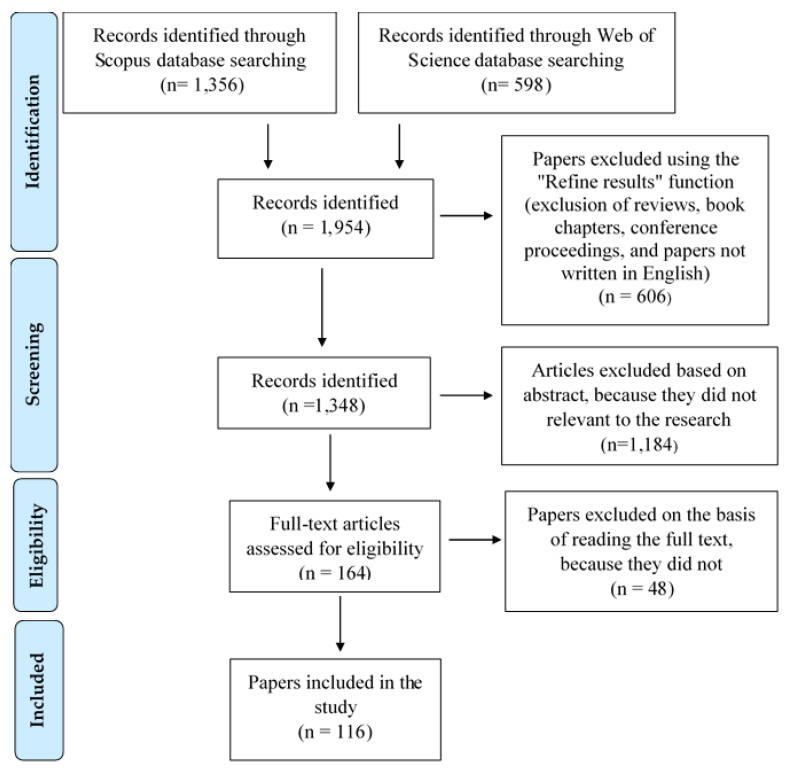
Methodological steps of the literature search process using a PRISMA flow diagram.

**Figure 2 foods-13-01092-f002:**
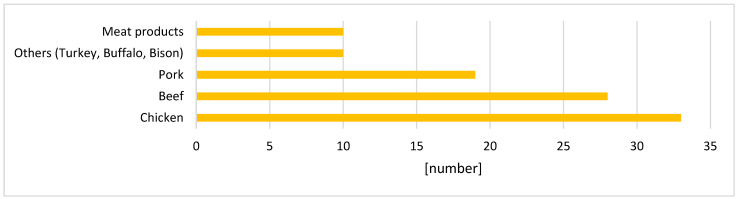
Meat types, subject to the innovations.

**Figure 3 foods-13-01092-f003:**
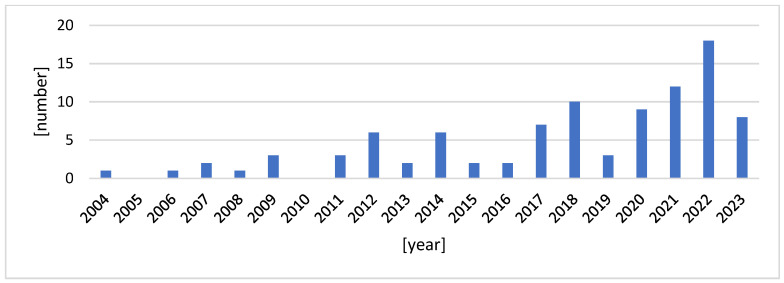
Number of papers per year related to the innovations (2004–2023).

**Table 1 foods-13-01092-t001:** Queries used in the database search.

Search String	Database
(“food” OR “agr*” AND “meat” AND “packaging” AND “consum*” OR “innovati*” OR “consum* behav*” OR “consum* preference*” OR “consum* attitude” OR “consum* concern*” OR “consum* intention*”)	Scopus
((“food” OR “agr*”) AND “meat” AND “packaging” AND (“consum*” OR “innovati*” OR “consum* behav*” OR “consum* preference*” OR “consum* attitude” OR “consum* concern*” OR “consum* intention*”))	Web of Science

**Table 2 foods-13-01092-t002:** Methodological approaches used in the selected studies.

**Acceptance scale**	Wang K et al., 2022 [89]; Guzek et al., 2020 [90]; Johnson et al., 2004 [91]; Mauricio et al., 2022 [92]; Polkinghorne et al., 2018 [93]; Van Wezemael et al., 2011 [94]
**Choice experiment and Willingness-to-Pay (WTP)**	Wang H et al., 2018 [71]; Grebitus et al., (a) 2013 [27]; Grebitus et al., (b) 2013 [95]; Chen et al., 2013 [96]; Ardeshiri et al., 2019 [97]; Erdem, 2015 [98]; Nocella et al., 2022 [99]
**Simultaneous Equation Model**	Sodano et al., 2016 [100]
**Consensus analysis/Qualitative study**	Chamorro et al., 2012 [101]; Horrillo et al., 2022 [102]; Pennanen et al., 2015 [103]; Htun et al., 2023 [18]; Barone et al., 2022 [88]
**Correspondence** **analysis**	Stoma et al., 2020 [104]

**Table 3 foods-13-01092-t003:** Data collection methods used in the selected studies.

Questionnaire	Sensory Evaluation	Focus Group
Wang K et al., 2022 [89]	Polkinghorne et al., 2018 [93]	Chamorro et al., 2012 [101]
Guzek et al., 2020 [90]	Johnson et al., 2004 [91]	Horrillo et al., 2022 [102]
Wang H et al., 2018 [71]		Ardeshiri et al., 2019 [97]
Sodano et al., 2016 [100]		Pennanen et al., 2015 [103]
Mauricio et al., 2022 [92]		Htun et al., 2023 [18]
Van Wezemael et al., 2011 [94]		Barone et al., 2022 [88]
Grebitus et al., 2013(a) [27]		
Grebitus et al., 2013(b) [95]		
Chen et al., 2013 [96]		
Erdem, 2015 [98]		
Stoma M et al., 2022 [104]		
Nocella et al., 2022 [99]		

**Table 4 foods-13-01092-t004:** Papers that investigated consumers’ preferences for innovations regarding fresh meat.

Source	Year	Country	Type of Meat	Innovation
Wang K. [89]	2022	China	Meat	Thermal pasteurization (TP) High-pressure processing (HPP) Irradiation (IR) Bacteriophages (BPs) Antimicrobial packaging (AP) Pulsed Electric Fields (PEFs) Rinsing meat carcasses with antimicrobial solutions (RMCA)
Stoma et al. [104]	2022	Poland	Food	Active and Intelligent packaging
Guzek et al. [90]	2020	Poland	Meat products	Novel packaging
Chamorro et al. [101]	2012	Spain	Meat	Fresh cuts Tray packaging in air Shrink-wrapped Controlled atmosphere packaging Modified atmosphere packaging Active packaging Intelligent packaging Frozen
Van Wezemael et al. [94]	2011	France, Germany, Poland, Spain, the UK	Beef	Modified atmosphere packaging Vacuum packaging Packaging containing protective bacteria Packaging releasing preservative food additives Packaging containing natural agents
Sodano et al. [100]	2016	Italy	Meat	Antimicrobial food packaging for meat and other foods
Ardeshiri et al. [97]	2019	The USA	Beef	Vacuum packaging
Polkinghorne et al. [93]	2018	Australia	Beef	Overwrap packaging using an oxygen-permeable film Vacuum skin packaging Modified atmosphere packaging
Wang H. et al. [71]	2018	China	Pork	Unpacked Plastic packed Vacuum sealed
Grebitus et al. (a) [27]	2013	The USA and Germany	Beef	Modified atmosphere packaging Modified atmosphere packaging with carbon monoxide
Grebitus et al. (b) [95]	2013	The USA	Beef	Modified atmosphere packaging Modified atmosphere packaging with carbon monoxide
Chen et al. [96]	2013	Canada	Beef	Vacuum packaging
Erdem [98]	2015	The UK	Chicken	Nanosensors
Johnson et al. [91]	2004	The USA	Chicken meat	Electron beam irradiation
Horrillo et al. [102]	2022	Spain	Lamb	Cherries and pecans
Mauricio et al. [92]	2022	Brazil	Lamb	Edible coating chitosan

**Table 5 foods-13-01092-t005:** Papers related to the study of consumers’ perception concerning innovations in intelligent packaging for fresh meat.

Source	Year	Country	Innovation
Nocella et al. [99]	2022	The UK	Smart biosensors
Barone et al. [88]	2022	The UK	Smart labels
Htun et al. [18]	2023	Finland, Spain, Poland, and Iceland	Smart tag packaging technologies
Pennanen et al. [103]	2015	Finland, Greece, France, and Germany	Time–temperature indicators

## Data Availability

The original contributions presented in the study are included in the article, further inquiries can be directed to the corresponding author.

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
