# Peer review of "Balancing Freshness and Sustainability: Charting a Course for Meat Industry Innovation and Consumer Acceptance"

_foods, 2024, doi:10.3390/foods13071092_

Round 1

Reviewer 1 Report

Comments and Suggestions for Authors

In this manuscript, the authors summarized and discussed the main technological innovations in the fresh meat industry and consumers' perceptions and acceptance of innovations designed to extend the shelf life of fresh meat. On this basis, they further investigated consumers' acceptance of innovations related to increasing shelf life, innovations related smart labels, drivers related to consumers’ acceptance and the existing barriers that hinder the complete embracement of these innovations. This work provides new insight and opinion into the development of innovative packaging for fresh pork. The manuscript is well-organized and clearly stated.

1. The introduction describes the sources of foodborne diseases and the serious phenomenon of food waste, but does not indicate whether meat products are a source of foodborne diseases or the proportion of meat products in food waste, which is insufficient to explain the importance of extending the shelf life of meat products.

2. It is suggested to add the author's name before the reference [7] quoted in page 2 line 75 to make the content of the article more fluent.

3. There are formatting problems in the article, such as lines 117, 272, 226 are left blank, and lines 203, 267, 297, 354 are not preceded by a space. It is suggested to unify the format of the article content to make the article look more organized.

4. Punctuation errors, such as missing punctuation after line 216 [53], line 229 [69], line 533[95] and multiple commas before "and" in line 489.

5. The font in figure1 on Page 4 is too small and the first box in the “Eligibility” row is offset.

6. At the beginning of 4.2 on page 5,“technological innovations” is divided into three parts. However, the font of "c)intelligent packaging" is different from others, and the later explanation section talks about “smart packaging” instead of “intelligent packaging”.

7. There is a space between two sentences in line 345.

8. The abbreviation of "thermal pasteurisation" in line 379 on page 10 is “(TP)” while it is “(PT)” in table 4 and in lines 384, 389, etc.

9. The reference "[81]" in line 530 on page 13 shows error, and the reference position of "[89]" in line 549 on page 14 is incorrect,which should be behind “study” instead of the author’s name.

10. There is a title “Limitation of study” in line 613 without a label.

Comments on the Quality of English Language

The language expression is relatively smooth, but it is still recommended to carefully read the entire text again and check the accuracy of the language

Author Response

  1. The introduction describes the sources of foodborne diseases and the serious phenomenon of food waste, but does not indicate whether meat products are a source of foodborne diseases or the proportion of meat products in food waste, which is insufficient to explain the importance of extending the shelf life of meat products.

Authors’ response: Thanks for your useful suggestion, the following description was added in the introduction: “Concerns about meat food safety stem from changes in animal production and distribution, increased international trade, and consumer preference for semi-processed products [6, 7]. Often, improper management leads to food contamination of meat or meat products, which can occur during distribution, processing, catering, or retailing [8] Such contamination can lead to the spread of foodborne outbreaks, with meat and meat products being a major source [9]. The consequences of even one case of a foodborne outbreak are significant and can have far-reaching impacts on businesses and loss of life [10]. In Warmate's study [11], foodborne illnesses are reported with data form the United States alone showing that, from 2007 to 2012, 163 outbreaks were identified, associated with 4132 illnesses, 772 hospitalizations, and 19 deaths associated with meat and poultry.”

  1. It is suggested to add the author's name before the reference [7] quoted in page 2 line 75 to make the content of the article more fluent.

Authors’ response: Thanks for the suggestion, we have added the reference required by referee.

  1. There are formatting problems in the article, such as lines 117, 272, 226 are left blank, and lines 203, 267, 297, 354 are not preceded by a space. It is suggested to unify the format of the article content to make the article look more organized.

Authors’ response: Thank you for the suggestion, we will make this change in the final version of the manuscript.

  1. Punctuation errors, such as missing punctuation after line 216 [53], line 229 [69], line 533[95] and multiple commas before "and" in line 489.

Authors’ response: Thank you for the suggestion we modified the text, in the final version of the manuscript.

  1. The font in figure1 on Page 4 is too small and the first box in the “Eligibility” row is offset.

Authors’ response: Thanks for the suggestion we enlarged the font.

  1. At the beginning of 4.2 on page 5,“technological innovations” is divided into three parts. However, the font of "c )intelligent packaging" is different from others, and the later explanation section talks about “smart packaging” instead of “intelligent packaging”.

Authors’ response: Thanks for the suggestion we have made the correction.

  1. There is a space between two sentences in line 345.

Authors’ response: Thank you for the suggestion we have made some changes, the others we will make on the final version of the manuscript.

  1. The abbreviation of "thermal pasteurisation" in line 379 on page 10 is “(TP)” while it is “(PT)” in table 4 and in lines 384, 389, etc.

Authors’ response: Thanks for the suggestion we have made the correction.

  1. The reference "[81]" in line 530 on page 13 shows error, and the reference position of "[89]" in line 549 on page 14 is incorrect,which should be behind “study” instead of the author’s name.

Authors’ response: Thanks for the suggestion we have made the correction.

  1. There is a title “Limitation of study” in line 613 without a label.

Authors’ response: Thanks for the suggestion we have made the correction.

Reviewer 2 Report

Comments and Suggestions for Authors

See in Annex, please.

Comments on the Quality of English Language

The English language is good.

Author Response

-P1-2, Introduction – The review on consumers’ perception, preferences, and acceptance  is highlighted in this manuscript. In the paragraph on P2, L68-70, the Authors should explain specifically why this issue deserves being investigated and reviewed.

Authors’ response: Thank you for your suggestion, we have added to the introduction a section as follows:

Concerns about meat food safety are due to changes in animal production and distribution, increased international trade, and consumer preference for semi-processed products (Shang and Tonsor, 2017 Lianou et al., 2017). Often, incorrect management leads to food contamination of meat or meat products, which can occur during distribution, processing, catering, or retailing (FSA, 2020). Such contamination can lead to the spread of foodborne outbreaks, of which meat and meat products are a major source (Nørrung et al., 2009). The consequences of even one case of a foodborne outbreak are very serious and can have great repercussions on businesses and loss of life (Hussain and Dawson, 2013). In Warmate's study (2023), foodborne illnesses are reported and it is stated that in the United States alone, from 2007 to 2012, 163 outbreaks were identified, associated with 4132 illnesses, 772 hospitalizations, and 19 deaths associated with meat and poultry.

In addition ,we also added the following part to justify the novelty of our research:

“To the best knowledge of the authors this paper presents a pioneering literature review that uniquely combines both technological and economic aspects. It offers a comprehen-sive exploration of how the meat industry can navigate the delicate balance between en-suring freshness and promoting sustainability, all while considering consumer ac-ceptance. Through a multidimensional approach, this review sheds light on the intricate interplay between technological advancements and economic viability within the meat industry landscape.

-L74-75 – “Until a few years ago, the terms “active” and “intelligent packaging” were used interchangeably” - Are the Authors sure of this affirmation? A reference should be included.

Authors’ response: Thanks for the suggestion, we have included the following reference:
Versino, F., Ortega, F., Monroy, Y., Rivero, S., López, O. V., & García, M. A. (2023). Sustainable and bio-based food packaging: A review on past and current design innovations. Foods12(5), 1057.

-P4, Figure 1 - The size of the letters should be increased.

Authors’ response: Thanks for the suggestion, we have increased the size of the figure

-P6, L220-221 – Revise this sentence, please. It is not clear.

Authors’ response: Thanks for the suggestion, we have modified the sentence as follows:
In addition, the use of radiant energy within some limitations does not change the main sensory characteristics of meat, such as color, flavor and texture.

-P6, L235 – On L230-235, the Authors cited some types of indicators. The citation of  sensors is lacking. The Authors certainly found references on this subject. Please, include in the text.

Authors’ response: Thanks for the suggestion, we have included the following reference:
Rodrigues, C., Souza, V. G. L., Coelhoso, I., & Fernando, A. L. (2021). Bio-based sensors for smart food packaging—Current applications and future trends. Sensors21(6), 2148. https://doi.org/10.3390/s21062148

-P8, Table 4 – Instead of “Cina”, use “China”.

Authors’ response: Thanks for the suggestion, we have edited the word

-P8, Table 4 – In the column “Innovation”, items should be cited following the same rule.

Authors’ response: Thanks for the suggestion we have made the correction.

-P9, L306-309 – A reference is lacking. Please, include a reference.

Authors’ response: The reference is reported in the first part of the paragraph, we have joined the sentences, avoiding the full stop in the final version of the manuscript.

-P9, L310-314 - A reference is lacking. Please, include a reference.

Authors’ response: The reference is reported in the first part of the paragraph, we have joined the sentences, avoiding the full stop in the final version of the manuscript.

-P10, L345-347 – A reference was cited [75], but the results are lacking.

Authors’ response: The reference is reported in the first part of the paragraph, we have joined the sentences, avoiding the full stop in the final version of the manuscript.

-P11, L390 – Include the reference.

Authors’ response: The reference is reported in the first part of the paragraph, we have joined the sentences, avoiding the full stop in the final version of the manuscript.

-P12, L434 – Include the reference.

Authors’ response: The reference is reported in the first part of the paragraph, we have joined the sentences, avoiding the full stop in the final version of the manuscript.

-P13, L486-497 – Should these paragraphs have references, or did they result from the  Authors’ observation

Authors’ response: The reference is reported in the first part of the paragraph, we have joined the sentences, avoiding the full stop in the final version of the manuscript.

Reviewer 3 Report

Comments and Suggestions for Authors

The manuscript BALANCING FRESHNESS AND SUSTAINABILITY: CHARTING A COURSE FOR MEAT INDUSTRY INNOVATION AND CONSUMER ACCEPTANCE, is innovative and contains relevant information, but minor details must be corrected , detailed below.

 Lines 30 – 32. I consider that this context should be referenced.

Lines 41 – 44. An overview of recent years generates more impact, the authors could mention how this problem has been growing in the last 5, 10 or 20 years.

Lines 46 – 48. Please cite examples of the methodologies.

Figure 3. Placing the years in increasing order is more convenient.

Lines 179 – 181. Discuss why it is relevant to mention the countries, what importance they have regarding the production or consumption of meat.

Lines 198 – 217. In addition to mentioning the natural sources of additives, it is important to highlight the main findings, or highlight which had a better technological and/or sensory response.

Consumers' acceptance section. As described in the previous observation, it is necessary to describe and discuss the main findings of the studies cited in Tables 2 and 3.

Line 530. There is an error phrase on this line.

Author Response

-Lines 30 – 32. I consider that this context should be referenced.

Authors’ response: R. Thanks for the suggestion, we have included the following reference: Echegaray, N., Hassoun, A., Jagtap, S., Tetteh-Caesar, M., Kumar, M., Tomasevic, I., ... & Lorenzo, J. M. (2022). Meat 4.0: principles and applications of industry 4.0 technologies in the meat industry. Applied Sciences, 12(14), 6986.

-Lines 41 – 44. An overview of recent years generates more impact, the authors could mention how this problem has been growing in the last 5, 10 or 20 years.

Authors’ response: Thanks for the suggestion. We pointed out in the introduction that the amount of food wasted annually is steadily increasing worldwide and is not limited only to industrialized countries (Wani et al., 2017).

Lines 46 – 48. Please cite examples of the methodologies.

Authors’ response: Thanks for the suggestion, we have added a reference related to the following statement: ..such as smart labels or packaging with the release of additives that allow the shelf life of products to be increased… , namely: Drago E, Campardelli R, Pettinato M, Perego P. Innovations in Smart Packaging Concepts for Food: An Extensive Review. Foods. 2020; 9(11):1628. https://doi.org/10.3390/foods9111628

Figure 3. Placing the years in increasing order is more convenient.

Authors’ response: Thanks for the suggestion, the figure has been modified as suggested by referee.

Lines 179 – 181. Discuss why it is relevant to mention the countries, what importance they have regarding the production or consumption of meat.

Authors’ response: Thanks for the suggestion, In almost all of these countries as previously reported, the amount of meat produced is increasing. We pointed out this aspect in the results and discussion (section 4.1)

-Lines 198 – 217. In addition to mentioning the natural sources of additives, it is important to highlight the main findings, or highlight which had a better technological and/or sensory response.

Authors’ response: Thanks for the suggestion, we have added the following text:

The antimicrobial action is highlighted by several studies. Dirpan et al. [15] demon-strated the antimicrobial effects of adding 10-15% garlic extract to beef packaging, while Tsironi et al. [47] explored the use of ginger and rosemary essential oils on lamb. Antibac-terial action was observed by Ali et al. [42] through the use of pectin and chitosan-based nanoactive films against multidrug-resistant meat pathogens such as Salmonella enterica, Escherichia coli, and Listeria monocytogenes. Contini et al. [43] reported excellent results against Escherichia coli and Staphylococcus aureus using chitosan films activated with lemongrass essential oil, and Nazari et al. [57] incorporated essential oils of Ziziphora clinopodioides into the film. Sganzerla et al. [58] demonstrated efficacy against Escherich-ia coli, Salmonella, and Shigella using starch, citric acid, and functionalized pectin with feijoa, while Hu X et al. [59] utilized an edible cellulose film activated with lysozymes. Tan et al. [60] showed that adding Chrysanthemum morifolium to a 1.5 percent (w/v) chitosan film increased the shelf life of raw meat products by significantly reducing Staphylococ-cus aureus activity. Additionally, antioxidant activity was observed in studies by Guo et al. [49], who incorporated carboxylated cellulose nanocrystals into films, and Ahmad et al. [61], who treated meat with kinnow and apple peel powder.

Consumers' acceptance section. As described in the previous observation, it is necessary to describe and discuss the main findings of the studies cited in Tables 2 and 3.

Authors’ response: Thanks for the suggestion, through a more in-depth analysis we have added the following statement:

Upon analysing the literature, interesting results emerge from a comprehensive ex-ploration of the methodologies adopted in consumer perception studies, encompassing both quantitative and qualitative approaches. This analysis sheds light on the limitations or biases that could impact the interpretation of results or the validity of any approaches.Regarding choice experiment-based methodologies, Ardeshiri et al. [98] report that the limited graphical space for the attributes analyzed in the choice experiment task may represent a restriction, as individuals may not evaluate all information when making their decision. Similarly, limitations are apparent in the study by Wang et al. [72], where the random draw method used to generate different sets of choices may be optimal according to various efficiency criteria, but may not offer the highest efficiency compared to the D-optimal or fractional factorial design. Other limitations were identified in the study by Er-dem [99], who conducted a discrete choice experiment and analyzed stated preference choice using a multinomial logit model. This study revealed that a limitation of the ana-lyzed model is the assumption of homogeneous preferences for all respondents. Conversely, Chen et al. [97] use a non-hypothetical choice experiment to isolate the impact of individual product characteristics on price and consumer choice decisions.  Such choice experiments are preferable to hypothetical experiments as they mitigate the potential hypothetical bias in participants' choice decisions. In order to report as objective results as possible, Wang et al. [90] eliminated ques-tionnaires completed in less than one third of the average time and those containing seven or more identical answers. In fact, improbable response patterns and sub-standard re-sponse times are signs of low quality responses. As for studies investigating consensus based on qualitative analysis, no particular limitations can be observed. From an operational point of view, many works conduct the investigation of consumer perceptions in several stages. In the study by Pennanen et al. [104], the operational phase is divided into three parts. The first part examines respond-ents' knowledge about the meat product and the type of packaging, the second part pro-vides participants with information about the Time-Temperature Indicator (TTI) concept, and a third part in which the two commercially available TTI applications (TTI1 and TTI2) are presented and discussed. Barone et al. [89] also divide the study into three phases. In the first phase, they ana-lyze consumers' general perceptions of freshness and expiry dates, as well as the purchas-ing, usage, storage, and disposal habits of red meat. The second phase analyzes consum-ers' perceptions of smart labels and their effect on food handling practices. Finally, the third phase explores consumer perceptions of smart labels and how these could be inte-grated and influence consumer practices in relation to the use, purchase, storage, and dis-posal of red meat. Htun et al. [18] also divided the study into four phases. In the first phase, the partici-pants were informed about the project together with other general information. In the sec-ond phase, the needs and desires of the participants were explored based on the features and benefits of the information transmitted by the smart tags, and finally, in the third phase, the acceptability of different smart tag solutions was investigated.

As for the second request, despite there is little information on demographic factors influence acceptance of different technologies we included additional literature in the section Drivers related to consumers' acceptance as follows:

Several recurring factors have been identified as key drivers in the various studies that examined consumers’ acceptance of innovative and sustainable solutions designed to extend the shelf life of fresh meat. These factors included gender, household composition, income, age, education level, place of residence, and product price.Regarding the perception of smart labels, no major gender differences were found in the study by Barone et al. [89]. However, in Pennanen's study, conducted in France and Germany [104], it emerged that female participants associated more benefits with Time-Temperature Indicators (TTI) than men. This difference was not found in Greece and Fin-land. Another important socio-economic aspect, highlighted in two different studies con-ducted out in Greece and Finland, was the composition of the household. In Greece, re-spondents belonging to households with children perceived the TTI as more beneficial than those belonging to single households, probably due to food security reasons. In Fin-land, respondents from households with other adults differed in perceiving the TTI as more advantageous, showing a higher preference among consumers with a lower average age.

  1. Line 530. There is an error phrase on this line.

Authors’ response: Thanks for the suggestion we have made the correction.

Reviewer 4 Report

Comments and Suggestions for Authors

1. The introduction could be more impactful by directly stating the research gaps earlier and succinctly summarizing the key contributions of the paper.

2. The review of technological innovations is thorough, yet the manuscript would benefit from a more explicit emphasis on the novelty and potential impact of the most recent advancements

3. Lines 236-413: The discussion on consumer perceptions is insightful but could be deepened by incorporating a more nuanced analysis of how demographic factors influence acceptance of different technologies . Discussing the implications of these findings for industry and policy-makers in more detail would also be valuable.

4. The discussion on the integration of technological innovations with consumer acceptance is insightful. However, the manuscript could benefit from a more detailed exploration of the methodologies used in consumer perception studies, including any limitations or biases that may affect the interpretation of results.

5. All references are correctly formatted and consistently applied throughout the manuscript, particularly in the Reference section

Comments on the Quality of English Language

1. Lines 7-10 and 27-30: the sentence could be streamlined for better clarity.

2. Ensuring consistency in terminology (e.g., "smart packaging" vs. "intelligent packaging") would improve the manuscript's professionalism.

Author Response

  1. The introduction could be more impactful by directly stating the research gaps earlier and succinctly summarizing the key contributions of the paper.

Authors’ response: Thank you for the feedback. Following the reviewer's recommendation, we have included more details about the gap in the literature that this work aims to address.

  1. The review of technological innovations is thorough, yet the manuscript would benefit from a more explicit emphasis on the novelty and potential impact of the most recent advancements

Authors’ response: Thanks for the suggestion, we have added the following section:

The antimicrobial action is highlighted by several studies. Dirpan et al. [15] demon-strated the antimicrobial effects of adding 10-15% garlic extract to beef packaging, while Tsironi et al. [47] explored the use of ginger and rosemary essential oils on lamb. Antibac-terial action was observed by Ali et al. [42] through the use of pectin and chitosan-based nanoactive films against multidrug-resistant meat pathogens such as Salmonella enterica, Escherichia coli, and Listeria monocytogenes. Contini et al. [43] reported excellent results against Escherichia coli and Staphylococcus aureus using chitosan films activated with lemongrass essential oil, and Nazari et al. [57] incorporated essential oils of Ziziphora clinopodioides into the film. Sganzerla et al. [58] demonstrated efficacy against Escherich-ia coli, Salmonella, and Shigella using starch, citric acid, and functionalized pectin with feijoa, while Hu X et al. [59] utilized an edible cellulose film activated with lysozymes. Tan et al. [60] showed that adding Chrysanthemum morifolium to a 1.5 percent (w/v) chitosan film increased the shelf life of raw meat products by significantly reducing Staphylococ-cus aureus activity. Additionally, antioxidant activity was observed in studies by Guo et al. [49], who incorporated carboxylated cellulose nanocrystals into films, and Ahmad et al. [61], who treated meat with kinnow and apple peel powder

  1. Lines 236-413: The discussion on consumer perceptions is insightful but could be deepened by incorporating a more nuanced analysis of how demographic factors influence acceptance of different technologies. Discussing the implications of these findings for industry and policy-makers in more detail would also be valuable.

Authors’ response: Thanks for the suggestion, despite there is little information on demographic factors influence acceptance of different technologies we included additional literature in the section Drivers related to consumers' acceptance as follows:

Several recurring factors have been identified as key drivers in the various studies that examined consumers’ acceptance of innovative and sustainable solutions designed to extend the shelf life of fresh meat. These factors included gender, household composition, income, age, education level, place of residence, and product price. Regarding the perception of smart labels, no major gender differences were found in the study by Barone et al. [89]. However, in Pennanen's study, conducted in France and Germany [104], it emerged that female participants associated more benefits with Time-Temperature Indicators (TTI) than men. This difference was not found in Greece and Fin-land.  Another important socio-economic aspect, highlighted in two different studies con-ducted out in Greece and Finland, was the composition of the household. In Greece, re-spondents belonging to households with children perceived the TTI as more beneficial than those belonging to single households, probably due to food security reasons. In Fin-land, respondents from households with other adults differed in perceiving the TTI as more advantageous, showing a higher preference among consumers with a lower average age.

Regarding the implications we have added the following paragraph:

Many of the participants in the reported surveys showed a lack of knowledge and/or inter-est in information about traceability and the benefits of healthy eating, which could be a significant barrier to the development of the innovative solutions covered in this study. Overcoming this barrier could be done through consumer education through the joint ac-tion of retailers and policy makers [29].  From an environmental sustainability perspective, active packaging is found to be more sustainable than conventional packaging, with lower rates of harm with respect to human health, ecosystems and resources [117]. This condition is fully in line with European and global policies that are increasingly focused on environmental and social sustainability of processes and products. Additionally, consumers are becoming more attentive to reducing environmental impact and mitigating the effects of climate change [118, 119]. Therefore, a thoughtful implementation of these concepts could further motivate consumers to pur-chase products featuring innovations such as active packaging or intelligent labels.

  1. The discussion on the integration of technological innovations with consumer acceptance is insightful. However, the manuscript could benefit from a more detailed exploration of the methodologies used in consumer perception studies, including any limitations or biases that may affect the interpretation of results.

Authors’ response: Thanks for the suggestion, through a more in-depth analysis we have added the following statement:

Upon analysing the literature, interesting results emerge from a comprehensive ex-ploration of the methodologies adopted in consumer perception studies, encompassing both quantitative and qualitative approaches. This analysis sheds light on the limitations or biases that could impact the interpretation of results or the validity of any approaches. Regarding choice experiment-based methodologies, Ardeshiri et al. [98] report that the limited graphical space for the attributes analyzed in the choice experiment task may represent a restriction, as individuals may not evaluate all information when making their decision. Similarly, limitations are apparent in the study by Wang et al. [72], where the random draw method used to generate different sets of choices may be optimal according to various efficiency criteria, but may not offer the highest efficiency compared to the D-optimal or fractional factorial design. Other limitations were identified in the study by Er-dem [99], who conducted a discrete choice experiment and analyzed stated preference choice using a multinomial logit model. This study revealed that a limitation of the ana-lyzed model is the assumption of homogeneous preferences for all respondents. Conversely, Chen et al. [97] use a non-hypothetical choice experiment to isolate the impact of individual product characteristics on price and consumer choice decisions.  Such choice experiments are preferable to hypothetical experiments as they mitigate the potential hypothetical bias in participants' choice decisions. In order to report as objective results as possible, Wang et al. [90] eliminated ques-tionnaires completed in less than one third of the average time and those containing seven or more identical answers. In fact, improbable response patterns and sub-standard re-sponse times are signs of low quality responses.  As for studies investigating consensus based on qualitative analysis, no particular limitations can be observed. From an operational point of view, many works conduct the investigation of consumer perceptions in several stages. In the study by Pennanen et al. [104], the operational phase is divided into three parts. The first part examines respond-ents' knowledge about the meat product and the type of packaging, the second part pro-vides participants with information about the Time-Temperature Indicator (TTI) concept, and a third part in which the two commercially available TTI applications (TTI1 and TTI2) are presented and discussed. Barone et al. [89] also divide the study into three phases. In the first phase, they ana-lyze consumers' general perceptions of freshness and expiry dates, as well as the purchas-ing, usage, storage, and disposal habits of red meat. The second phase analyzes consum-ers' perceptions of smart labels and their effect on food handling practices. Finally, the third phase explores consumer perceptions of smart labels and how these could be inte-grated and influence consumer practices in relation to the use, purchase, storage, and dis-posal of red meat. Htun et al. [18] also divided the study into four phases. In the first phase, the partici-pants were informed about the project together with other general information. In the sec-ond phase, the needs and desires of the participants were explored based on the features and benefits of the information transmitted by the smart tags, and finally, in the third phase, the acceptability of different smart tag solutions was investigated.

  1. All references are correctly formatted and consistently applied throughout the manuscript, particularly in the Reference section

Comments on the Quality of English Language

  1. Lines 7-10 and 27-30: the sentence could be streamlined for better clarity.

Authors’ response: Thanks for the suggestion we have made the correction.

  1. Ensuring consistency in terminology (e.g., "smart packaging" vs. "intelligent packaging") would improve the manuscript's professionalism.

Authors’ response: Thanks for the suggestion we have made the correction.